# REASONING WITH CONFIDENCE: EFFICIENT VERIFICATION OF LLM REASONING STEPS VIA UNCERTAINTY HEADS

## ABSTRACT

Solving complex tasks usually requires LLMs to generate long multi-step reasoning chains. Previous work has shown that verifying the correctness of individual reasoning steps can further improve the performance and efficiency of LLMs on such tasks and enhance solution interpretability. However, existing verification approaches, such as Process Reward Models (PRMs), are either computationally expensive, limited to specific domains, or require large-scale human or model-generated annotations. Thus, we propose a lightweight alternative for step-level reasoning verification based on data-driven uncertainty scores. We train transformer-based uncertainty quantification heads (UHeads) that use the internal states of a frozen LLM to estimate the uncertainty of its reasoning steps during generation. The approach is fully automatic: target labels are generated either by another larger LLM (e.g., DeepSeek R1) or in a self-supervised manner by the original model itself. UHeads are both effective and lightweight, containing less than 10M parameters. Across multiple domains, including mathematics, planning, and general knowledge question answering, they match or even surpass the performance of PRMs that are up to 810× larger. Our findings suggest that the internal states of LLMs encode their uncertainty and can serve as reliable signals for reasoning verification, offering a promising direction toward scalable and generalizable introspective LLMs.

## 1 INTRODUCTION

Chain-of-thought (CoT) prompting has proven highly effective in eliciting the reasoning capabilities of large language models (LLMs) for solving complex tasks (Wei et al., 2022). Recent post-training approaches further enhance this ability through reinforcement learning, rewarding models for generating responses that conform to the CoT pattern and yielding correct final answers (DeepSeek-AI, 2025; Yang et al., 2025b; Abdin et al., 2025). However, the emphasis on verifying only final answers raises concerns about the reliability of intermediate reasoning steps (Lightman et al., 2023; Zheng et al., 2025; Barez et al., 2025). Flawed steps can propagate and systematically distort conclusions (Fu et al., 2025b), while in some cases LLMs may still produce the correct answer despite erroneous intermediate steps (Arcuschin et al., 2025). Such issues undermine trust in CoT-based methods, particularly in high-stakes domains like medicine (Amann et al., 2020) and law (Fan et al., 2025).

A common way to supervise models for correctness of reasoning steps is through process reward models (PRMs: Lightman et al. (2023); Zhang et al. (2025c)). However, PRM-based supervision faces key drawbacks. First, it requires deploying an additional LLM supervisor, adding substantial computational overhead. Second, it often relies on MC estimation for training data annotation, but this approach fails in tasks like mathematical proofs (Azerbayev et al., 2023) and planning (Zheng et al., 2024), where step-level correctness cannot be inferred from the final result alone.

Another promising line of work for assessing the correctness of reasoning steps leverages uncertainty quantification (UQ) methods (Gal et al., 2016; Malinin & Gales, 2021). Unlike PRMs, which rely on external verification, UQ assumes that a model's outputs and internal states provide information about the reliability of its generations. To date, only relatively simple unsupervised UQ methods have been investigated for LLM reasoning (Fu et al., 2025b; Yan et al., 2025). While com-

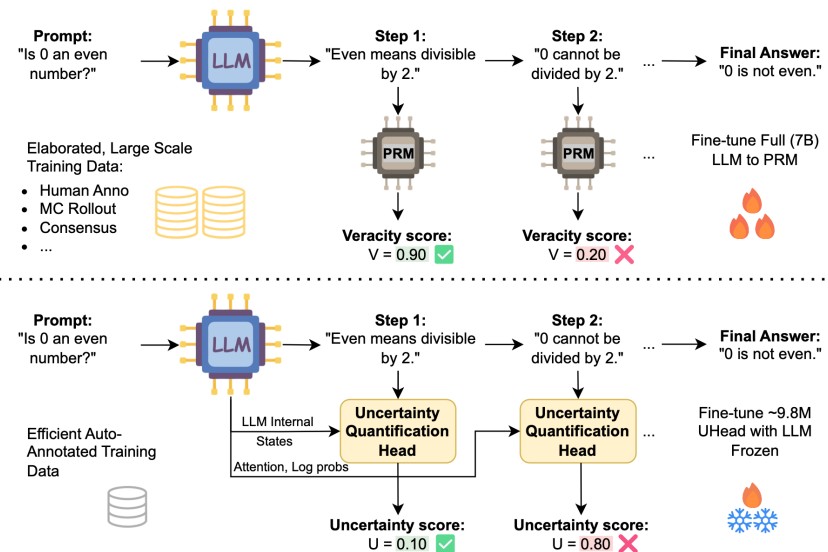

Figure 1: Process reward models (top) in comparison to uncertainty quantification heads (bottom).

putationally lightweight, these approaches consistently fail to capture subtle reasoning flaws and lag significantly behind PRMs in performance.

In this work, we ask: Can the cost-efficiency of UQ methods be combined with the performance of PRMs? Recent research in UQ shows that data-driven approaches substantially outperform unsupervised ones in detecting factual hallucinations (Azaria & Mitchell, 2023; Chuang et al., 2024; Shelmanov et al., 2025). Building on this insight, we introduce a lightweight data-driven uncertainty-based verifier for step-level reasoning. Specifically, we adapt the UHead framework (Shelmanov et al., 2025) to train uncertainty quantification heads on top of frozen LLMs for reasoning step verification. Unlike PRMs, which rely on generated text as input, UHeads exploit internal LLM states (attention and token probabilities) as features, making them far more efficient (see Figure 1): UHeads trained in our work require less than 10M parameters. Training labels are generated automatically, either by using a powerful external LLM (e.g., DeepSeek R1) as a judge or by relying on the original LLM itself. This enables scalable supervised or self-supervised training without the need for verifiable final answers, human annotations, or costly Monte Carlo rollouts.

Extensive in-domain (ID) and out-of-domain (OOD) evaluation highlights UHeads' advantages in three key aspects: (1) **Generalizable Performance**: UHeads achieve competitive or superior performance compared to PRM baselines in step-level verification and test-time scaling, including best-of-$N$ selection and step-level correctness supervision, especially when dealing with OOD reasoning tasks (e.g., reasoning-intensive QA and planning); (2) **Computational Efficiency**: UHeads substantially outperform PRMs that are 150× larger, while remaining competitive with PRMs up to 800× larger; and (3) **Training Data Efficiency**: unlike PRMs, which often rely on proprietary datasets (He et al., 2024b), costly human annotations (Lightman et al., 2023), or consensus-filtering pipelines involving both Monte Carlo rollouts and LLM judgments (Zhang et al., 2025c), UHeads can be trained on data annotated automatically in a self-supervised manner.

Our contributions are the following:

- We introduce a lightweight approach to verifying step-level correctness in LLM-generated reasoning by extending the UHead framework. Our 9.8M-parameter UHead achieves competitive or superior performance compared to 7B-scale PRMs across diverse reasoning tasks and settings, while being significantly more efficient at inference time in terms of memory and compute.
- We develop an efficient and scalable data annotation pipeline that eliminates the need for human-labeled data, verifiable final answers, or costly Monte Carlo rollouts. We further show

that high performance in reasoning verification can be achieved using annotations derived either from external verification models (e.g., DeepSeek R1) or from a fully self-supervised process.

- We show that our approach achieves strong generalization across domains: a single UHead trained only on mathematical tasks generalizes effectively to planning and general knowledge QA. Unlike PRMs, which often overfit to domain-specific reasoning patterns, our lightweight UHead verifier leverages internal LLM uncertainty signals, enabling robust reasoning verification without requiring domain adaptation.

## 2 BACKGROUND

### 2.1 TEST-TIME SCALING

State-of-the-art LLMs solve complex tasks by generating CoT reasoning traces (Guo et al., 2025). Given an input question $\mathbf{x}$, models produce a sequence of reasoning steps $\mathbf{r} = \{r_1, r_2, \dots, r_T\}$ followed by a final answer $\mathbf{y} = g(\mathbf{r}, \mathbf{x})$. Formally, an LLM parameterized by $\theta$ models the conditional distribution:

$$P_\theta(\mathbf{y}, \mathbf{r} \mid \mathbf{x}) = \prod_{t=1}^{T} P_\theta(r_t \mid \mathbf{x}, \mathbf{r}_{<t}) \cdot P_\theta(\mathbf{y} \mid \mathbf{x}, \mathbf{r}),$$

where $r_t$ denotes the $t$-th reasoning step and $\mathbf{r}_{<t}$ are the preceding steps. Although CoT reasoning improves final task performance, locally incorrect steps $r_t$ can propagate and cause wrong answers.

*Test-time scaling (TTS)* algorithms (Yao et al., 2023; Snell et al., 2024) aim to further improve the performance of LLMs without retraining the entire model, by allocating additional computation or using more sophisticated inference strategies. By leveraging multiple candidate solutions and verification signals, TTS facilitates selecting a correct answer. Common approaches include online and offline best-of-$N$ (BoN) sampling. Other techniques employ sophisticated search and sampling techniques based on beam search and backtracking, such as the tree of thoughts (Yao et al., 2023).

**Online best-of-$N$ sampling** (Yao et al., 2023) assumes that on each step, we generate $N$ continuations $\{r_t^{(1)}, r_t^{(2)}, \dots, r_t^{(N)}\}$, and each of them is evaluated online using a quality function $Q_{\text{online}}\left(r_{1:t}^{(j)}\right)$. In the greedy algorithm, only the most promising option is expanded further. After reaching the final step $T$, the answer is obtained from the greedily derived chain:

$$r_t^* = \arg\max_{j \in \{1, \dots, N\}} Q_{\text{online}}\left(\mathbf{r}_{1:t}^{(j)}\right), \ \mathbf{r}^* = \{r_1^*, r_2^*, \dots, r_T^*\}, \ \mathbf{y} = g(\mathbf{x}, \mathbf{r}^*).$$

**Offline best-of-$N$ sampling** (Wang et al., 2023) assumes that we sample not one, but $N$ reasoning chains $\{\mathbf{r}^{(1)}, \mathbf{r}^{(2)}, \dots, \mathbf{r}^{(N)}\}$. Each chain $\mathbf{r}^{(j)}$ is evaluated using a scoring function $Q_{\text{offline}}\left(\mathbf{r}^{(j)}\right)$ (see Section 2.2), and the final answer is derived from the best-scoring chain:

$$\mathbf{r}^* = \arg\max_{j \in \{1, \dots, N\}} Q_{\text{offline}}\left(\mathbf{r}^{(j)}\right), \quad \mathbf{y} = g(\mathbf{x}, \mathbf{r}^*).$$

### 2.2 PROCESS REWARD MODELS

For mathematical and logical reasoning, PRMs are a prominent approach for quality assessment in both online and offline test-time scaling (Lightman et al., 2023; Luo et al., 2024; Wang et al., 2024b). PRMs are critics designed to estimate the quality of each partial reasoning state $\mathbf{r}_{1:t}^{(j)}$ by assigning a reward $R_{\text{PRM}}\left(\mathbf{r}_{1:t}^{(j)}\right)$ that reflects the likelihood of the chain eventually leading to a correct solution. PRMs typically represent a separate LLM trained to evaluate the plausibility and correctness of intermediate reasoning steps. The annotation for training PRMs comes from various sources, including crowdsourcing, self-consistency checking, and synthetic data generation pipelines.

In online BoN, we simply use the reward from PRM as a step-wise quality function. In offline BoN, the score for the complete chain $\mathbf{r}^{(j)}$ can be obtained via a temporal aggregation of process rewards, e.g., minimum step score (Zhang et al., 2025c):

$$Q_{\text{online}}\left(\mathbf{r}_{1:t}^{(j)}\right) = R_{\text{PRM}}\left(\mathbf{r}_{1:t}^{(j)}\right), \quad Q_{\text{offline}}\left(\mathbf{r}^{(j)}\right) = \min_{1 \le t \le T^{(j)}} R_{\text{PRM}}\left(\mathbf{r}_{1:t}^{(j)}\right).$$

Although PRMs have proven to be highly effective even in boosting the performance of larger LLMs, they come with several limitations: they are relatively large models, typically containing 1.5B–8B parameters, which leads to additional computational and memory overhead during inference. Moreover, PRMs are often domain-specific, being trained for tasks such as mathematical reasoning, and exhibit limited generalization to unseen domains.

## 2.3 UNCERTAINTY QUANTIFICATION

UQ is a conceptual and methodological framework that bridges Bayesian and information-theoretical approaches to modeling (Maddox et al., 2019) with various practical tasks such as selective classification (Geifman & El-Yaniv, 2017), out-of-distribution detection (Mukhoti et al., 2023), and adversarial attack detection (Lee et al., 2018). The core assumption behind UQ is that ML models can produce not only predictions but also signals indicative of their reliability. In the context of LLMs, uncertainty reflects how likely a generated token, reasoning step, or the whole generated sequence is to be correct given the model's internal beliefs and available context. For classification tasks, total uncertainty for a model trained on data $\mathcal{D}$ could be represented as the entropy of the predictive posterior:

$$U(\mathbf{y}) = H\big[P(\mathbf{y} \mid \mathbf{x}, \mathcal{D})\big] = \mathbb{E}_{P(\theta \mid \mathcal{D})}\big[H\big[P_\theta(\mathbf{y} \mid \mathbf{x})\big]\big] + \mathbb{E}_{P(\theta \mid \mathcal{D})}\Big[\mathrm{KL}\big(P_\theta(\mathbf{y} \mid \mathbf{x}) \,\big\|\, P(\mathbf{y} \mid \mathbf{x}, \mathcal{D})\big)\Big].$$

In practice, it is common to use various heuristics and approximations, which can be of various natures. However, unlike external verification methods, UQ primarily leverages internal model capabilities to provide these estimates, such as logits, multiple sampled predictions, hidden states, and attention weights (see Figure 1).

UQ for LLMs poses unique challenges beyond standard text classification, making it difficult to establish an effective universal unsupervised solution (Zhang et al., 2023; Kuhn et al., 2023a; Duan et al., 2024; Fadeeva et al., 2024). These limitations have motivated the exploration of data-driven approaches, which have consistently demonstrated superior empirical performance (Azaria & Mitchell, 2023; He et al., 2024a; Chuang et al., 2024).

## 3 VERIFICATION OF REASONING STEPS WITH UNCERTAINTY QUANTIFICATION HEADS

**Method.** UQ offers an efficient alternative to PRMs for assessing the quality of reasoning steps during online test-time scaling. While PRMs leverage generated sequences, UQ methods leverage internal capabilities of LLMs. Instead of training large reward models, it is possible to use lightweight UQ methods or train lightweight data-driven uncertainty estimators:

$$Q_{\text{online}}\big(r_t^{(j)}\big) = 1 - U\big(r_t^{(j)} \mid r_{<t}^{(j)}, \mathbf{x}\big), \quad Q_{\text{offline}}\big(\mathbf{r}^{(j)}\big) = \min_{1 \le t \le T^{(j)}} \Big(1 - U\big(r_t^{(j)} \mid r_{<t}^{(j)}, \mathbf{x}\big)\Big).$$

We adopt a state-of-the-art data-driven method for uncertainty quantification (UQ) – *uncertainty quantification heads (UHeads)* (Shelmanov et al., 2025). UHeads are lightweight supervised auxiliary modules for LLMs, designed to estimate uncertainty and detect hallucinations. Implemented as compact transformer-based classifiers, they leverage the internal states of the base LLM without requiring fine-tuning or modification of the model or its outputs. Unlike PRMs, which are domain-specific (e.g., math proofs or planning) and incur significant additional inference costs, UHeads are *plug-and-play* modules that generalize across tasks, domains, and languages, making them practical for reasoning step assessment.

While PRMs rely solely on generated tokens, UHead leverages richer per-token features extracted from the base LLM's internal states, specifically, attention weights to the 1-3 preceding tokens and the logits of the top-K candidate generations. Its architecture is transformer-based: each per-token feature vector is first linearly projected and then processed through a stack of $L$ Transformer blocks to capture intra-step dependencies. The resulting token representations are mean-pooled across the reasoning step to obtain a step-level vector, which is passed through a two-layer classification head with dropout and a GeLU activation to produce the final class logits.

Originally developed to score atomic factual claims (Fadeeva et al., 2024), here we extend UHeads to score reasoning steps in a PRM-style setup.

**Constructing training dataset for UHead.** To construct the training data for UHead, we use 10.8K problems (prompts) from the PRM800K training dataset. This dataset is an established resource for training and evaluating LLM reasoning capabilities. As most PRMs are already trained on LLM generations derived from it, it enables a fair comparison (Wang et al., 2024a; Zhang et al., 2025c).

We prompted reasoning LLMs to generate each step of the CoT solution on a separate, self-contained line, enabling accurate correctness grading. The full prompt is provided in Appendix C.1. `Qwen3-8B` reliably follows this prompt and produces high-quality CoT steps across domains. For each of the 10.8K math problems, we generate 3 reasoning chains, resulting in ~32K data samples. For all reasoning chain generations, we use nucleus sampling with a maximum of 256 new tokens, top-$k$ = 50, top-$p$ = 0.95, and a temperature of 1.0.

Then, we annotated step-level correctness of each CoT step using an LLM as a judge. The judge LLM was provided with the question, the target LLM's CoT steps and final answer, and the ground-truth answer; and then prompted to assess the correctness of each step. The full annotation prompts are given in Appendix C.2. We consider two judging approaches: (1) an external verifier, where a larger LLM evaluates the steps, and (2) a self-supervised setting, where the same reasoning LLM annotates its own generations. Following Zheng et al. (2025), who showed that general reasoning models perform well at step-level correctness annotation, we adopt `DeepSeek-R1` as the external verifier. Other details to reproduce the training data can be found in Appendix D.1.

## 4 EXPERIMENTAL SETUP

### 4.1 LLMs FOR REASONING, UHEAD TRAINING DETAILS, EVALUATION DATASETS

**LLMs for reasoning.** We conduct experiments with two state-of-the-art LLMs, `Qwen3-8B` (Yang et al., 2025a) and `Phi-4` (Abdin et al., 2025). For these models, we prepared training datasets for reasoning-step verification and trained UHeads on them.

**UHead training settings.** We follow Shelmanov et al. (2025)'s recipe to train UHeads with a frozen underlying LLM. For all experiments with different training data and underlying LLMs, we use the same hyperparameter settings, which are detailed in Appendix D.2.

**Evaluation datasets** span three domains: mathematical reasoning (in-domain), planning (OOD), and general knowledge QA (OOD). We select test sets that demand non-trivial reasoning and include reasoning steps that are unambiguously verifiable, enabling reliable evaluation by both LLMs and human annotators. The details for test datasets are presented in Appendix E.

### 4.2 BASELINES

We benchmark against a broad set of baselines, covering PRMs of different sizes and UQ methods.

**Small PRMs (1.5B)** include 2 PRMs fine-tuned from Qwen2.5-Math-1.5B: `Skywork-PRM-1.5B` (He et al., 2024b) and `H4-Qwen2.5-PRM-1.5B-0.2` (HuggingFaceH4, 2025).

**Large PRMs (7–8B)** include (1) `Math-Shepherd-PRM-7B` (Wang et al., 2024b), which determines the process labels for each step by estimating the empirical probability of reaching the correct final answer (MC estimation); (2) `RLHFlow-PRM-8B-DeepSeek/Mistral` models (Xiong et al., 2024) trained with MC-estimated labels from DeepSeek/Mistral rollouts; (3) `Universal-PRM-7B` (AURORA, 2025) trained using ensemble prompting and reverse verification; (4) `Qwen2.5-Math-7B-PRM800k` trained on the PRM800k dataset (Lightman et al., 2023); and (5) `Qwen2.5-Math-PRM-7B` (Zhang et al., 2025c), which combines MC estimation with LLM-as-a-Judge consensus and currently achieves the best result on ProcessBench compared to PRMs of similar scale and computation (Zhao et al., 2025).

**UQ methods** evaluated in our experiments fall into two categories, differing in computational cost: (1) *lightweight scores that use only single generation*: `Maximum Sequence Probability`, `Mean Token Entropy`, `Perplexity` (Fadeeva et al., 2023), `P(True)` (Kadavath et al., 2022), `CCP` (Fadeeva et al., 2024), and `Self-Certainty` (Kang et al., 2025), which show significant advantages in reasoning chain selection (Fu et al., 2025b); and (2) *sampling-based scores*: `Semantic Entropy`, `Lexical Similarity` (Kuhn et al., 2023b), and `Degree Matrix`

(Lin et al., 2024). To compute the latter, we draw $M = 10$ alternative steps per step position, so they are much less computationally efficient. All methods are implemented using the LM-Polygraph framework (Fadeeva et al., 2023; Vashurin et al., 2025).

## 4.3 Evaluation Settings

We conduct experiments in three scenarios: (1) step-level correctness prediction; (2) offline best-of-$N$ selection; and (3) online best-of-$N$ selection.

**(1) Step-level correctness prediction.** For each question in the test set, we generate a CoT trace and evaluate step-level correctness using `DeepSeek-R1`. To improve the accuracy of its judgments, the model is provided with all metadata, such as the ground-truth answer, reasoning steps, and supporting evidence (see prompt in Appendix C.2). To validate the reliability of our evaluation pipeline, we evaluate it against (1) the human annotations from a random subset of PRM800k and (2) a manually annotated set of ~1000 steps spanning QA, planning, and ProofNet tasks (see Appendix G for annotation details). DeepSeek-R1 achieves 95% Acc. on PRM800k and ~90% on other datasets (see Table 1).

Table 1: Accuracy of the DeepSeek-R1-based step-level evaluation pipeline relative to human labels.

| Dataset | # Steps | Acc. |
|---|---|---|
| PRM800K | 17,067 | 95.29 |
| ProofNet | 193 | 87.05 |
| Trips | 102 | 93.06 |
| Meetings | 101 | 95.70 |
| Calendar | 102 | 91.09 |
| StrQA | 313 | 95.85 |
| SciQA | 245 | 99.28 |

**(2) Offline best-of-$N$.** In the Best-of-$N$ (BoN) setting, we generate $N$ reasoning chains per problem ($N$=10 for the math and QA datasets, $N$=5 for the planning datasets, temperature=1.0). We compute the quality score $Q$ for each step, and the chain with the highest score is selected for deriving the final answer. Correctness of the solution is measured using accuracy. For GSM8K, we use exact match against the gold-standard final answer. For other datasets, where final answers may be open-ended or structurally complex, we use DeepSeek-R1 to provide binary grades of the answers based on both the problem and the reference solution. The grading prompt is provided in Appendix C.2.

**(3) Online best-of-$N$** selects the best candidate at each step during the generation process. At each step, the system produces $N$ candidate steps, scores them, and expands the best option. This process repeats until an end-of-sequence token is reached, after which the chain is evaluated using the same metrics as in the offline BoN. This setting evaluates how well PRMs and UQ methods can guide the LLM reasoning trajectory. For computational efficiency, we pick $N$=5 and use a generation temperature of 1.5 to ensure the diversity of candidate steps.

## 5 Results and Analysis

**Step-level correctness prediction** results for Qwen-3 are presented in Table 2 and for Phi-4 in Table 6 in Appendix A. We conclude that *unsupervised UQ methods in many cases provide valuable signals for detecting reasoning errors*. Even lightweight single-sample techniques such as MSP and MaxEntropy demonstrate slight improvements over the random baseline in the majority of datasets, with the exception of calendar planning. Sampling-based methods such as Semantic Entropy usually perform slightly better, but have similar issues on calendar planning.

Small PRMs demonstrate notable improvements over unsupervised UQ methods for mathematical problems and QA, but for OOD planning datasets, they are not better than UQ. Best large PRMs considered in our work, such as Qwen2.5-Math-7B-PRM800k and Qwen2.5-Math-PRM-7B, substantially outperform all unsupervised UQ methods and smaller PRMs on all tasks.

Despite using 750-810× fewer parameters than PRMs, *both UHead variants perform on par with, or even surpass the best PRMs*. For ID mathematical datasets, UHeads substantially outperform all other PRM baselines except the two strongest PRMs based on Qwen2.5-Math. On MATH, UHead trained on self-supervised annotation is on par with Qwen2.5-Math-PRM-7B. On GSM8K, UHead trained on DeepSeek annotation is on par with Qwen2.5-Math-7B-PRM800k. For ProofNet, UHead falls behind the strongest Qwen-based PRMs, but substantially outperforms all others.

It is not surprising that parameter-heavy PRMs achieve the best results on ID mathematical datasets, as they tend to strongly overfit to this domain. In contrast, UHead, with far fewer parameters, avoids such domain-specific overfitting. This becomes evident on planning tasks, where UHead

Table 2: PR-AUC↑ for detecting incorrect reasoning steps (Qweb3-8B). Best scores are shown in **bold**. Other competitive scores show clear advantages are underlined. # Sample indicates the number of training samples; each sample corresponds to a reasoning trajectory with step-level labels.

| Method | # Sample | Math (ID) | | | Planning (OOD) | | | QA (OOD) | | Average | | |
|---|---|---|---|---|---|---|---|---|---|---|---|---|
| | | MATH | GSM8k | ProofNet | Trips | Meetings | Calendar | StrQA | SciQA | ID | OOD | Overall |
| *Unsupervised Uncertainty Quantification (UQ)* | | | | | | | | | | | | |
| Random | - | .173 | .061 | .153 | .524 | .588 | .486 | .116 | .125 | .129 | .368 | .278 |
| MaxProb | - | .221 | .106 | .194 | .578 | .655 | .483 | .114 | .259 | .174 | .418 | .326 |
| MaxEntropy | - | .212 | .122 | .185 | .545 | .618 | .443 | .119 | .227 | .173 | .390 | .309 |
| Perplexity | - | .205 | .099 | .176 | .519 | .572 | .418 | .110 | .228 | .160 | .369 | .291 |
| Self-Certainty | - | .213 | .101 | .155 | .516 | .643 | .482 | .120 | .243 | .156 | .407 | .309 |
| CCP | - | .250 | .090 | .168 | .584 | .645 | .452 | .119 | .235 | .169 | .407 | .318 |
| P(True) | - | .164 | .059 | .172 | .535 | .608 | .490 | .126 | .263 | .132 | .404 | .302 |
| Semantic Entropy | - | .257 | .116 | .173 | .565 | .610 | .492 | .111 | .265 | .182 | .409 | .324 |
| Lexical Similarity | - | .250 | .119 | .170 | .569 | .603 | .490 | .120 | .254 | .180 | .407 | .322 |
| Degree Matrix | - | .227 | .089 | .147 | .534 | .597 | .484 | .107 | .258 | .154 | .396 | .305 |
| *PRMs 150× Larger than UHeads* | | | | | | | | | | | | |
| Skywork-PRM-1.5B | Unk | .283 | .412 | .147 | .433 | .532 | .502 | .254 | .408 | .281 | .426 | .371 |
| H4-Qwen2.5-PRM-1.5B-0.2 | 369K | .259 | .171 | .159 | .597 | .633 | .495 | .213 | .228 | .196 | .433 | .344 |
| *PRMs 750× to 810× Larger than UHeads* | | | | | | | | | | | | |
| Math-Shepherd-PRM-7B | 440K | .380 | .405 | .147 | .662 | .660 | .657 | .284 | .415 | .311 | .536 | .451 |
| RLHFlow-PRM-Deepseek-8B | 253K | .289 | .540 | .136 | .583 | .579 | .504 | .390 | **.518** | .322 | .515 | .442 |
| RLHFlow-PRM-Mistral-8B | 273K | .233 | .537 | .118 | .523 | .555 | .499 | .349 | .415 | .296 | .468 | .404 |
| Universal-PRM-Qwen2.5-Math-7B | 690K | .534 | .624 | **.329** | .730 | .753 | .691 | .328 | .330 | .496 | .566 | .540 |
| Qwen2.5-Math-7B-PRM800k | 265K | **.586** | .613 | .301 | .708 | .768 | .727 | .362 | .404 | .500 | .594 | .559 |
| Qwen2.5-Math-PRM-7B | 860K | .531 | **.702** | .310 | .711 | .757 | .745 | .334 | .429 | **.514** | .595 | **.565** |
| *Uncertainty Heads (UHeads)* | | | | | | | | | | | | |
| ★ UHead Self-anno (Ours) | **32K** | .529 | .594 | .260 | .735 | .779 | .779 | .394 | .404 | .461 | **.618** | .559 |
| ★ UHead DeepSeek-anno (Ours) | **32K** | .465 | .616 | .243 | **.740** | **.802** | **.786** | **.395** | .361 | .441 | .617 | .551 |

consistently outperforms the strongest PRMs across all datasets. The best performance is obtained by UHead trained on DeepSeek annotations, though even the self-supervised variant surpasses all PRMs. For QA datasets, the picture is more mixed. On StrategyQA, UHead again emerges as the top step-verification method. However, on ScienceQA, RLHFlow-PRM-DeepSeek makes a clear leap forward, outperforming other techniques, including Qwen-based PRMs. Still, UHead performs on par with Qwen-based PRMs, underscoring its solid performance in this setting.

In summary, UHead lags behind the strongest PRMs on average for ID tasks but outperforms them for OOD tasks. Notably, the average performance of the self-supervised UHead is comparable to that of the externally supervised variant, highlighting its ability to provide an *efficient self-supervised solution for reasoning step verification*, particularly valuable in OOD settings. Similar results are observed for Phi-4.

**Offline best-of-$N$** results for Qwen-3 are presented in Table 3 (Phi-4 results are in Table 7, Appendix A). Both UHead types achieve the best offline BoN performance on several benchmarks, including GSM8K, ProofNet, Meeting/Calendar Planning, StrategyQA, and ScienceQA, achieving on par or superior performance to the best PRMs. On MATH and Trip Planning, UHeads rank second-best. Notably, UHead-based BoN selection enables Qwen3-8B to outperform its larger counterpart Qwen3-14B on multiple benchmarks (MATH, GSM8K, ProofNet, Meeting Planning, and ScienceQA).

As in the step-level setting, UHead exhibits strong generalization. While the best PRMs (Universal-PRM-7B, Qwen2.5-MATH-7B-PRM, and Qwen2.5-Math-PRM-7B) reach parity with UHead on in-domain datasets MATH and GSM8K, they often fall behind on OOD tasks, such as planning and StrategyQA. Furthermore, UHead provides stable gains across tasks of varying difficulty, from relatively simple datasets (MATH, GSM8K, ScienceQA) to complex planning benchmarks.

**Online best-of-$N$** evaluation results are presented in Table 2. While PRMs remain highly competitive on ID tasks like MATH and GSM8K, UHead trained on Deepseek-R1 annotations has obtained the runner-up accuracy across all three ID datasets. More impressively, UHeads demonstrate superior generalization to OOD. UHead DeepSeek-anno secures the highest overall and OOD average accuracy, outperforming all PRMs by a considerable margin. By excelling across both ID and OOD tasks with drastically fewer resources, UHeads proves to be a powerful substitute for PRMs as a test-time reasoning guidance tool.

Table 3: Offline best-of-$N$ decoding accuracy across datasets (Qwen-3 8B). For datasets with verifiable final answer, we also provide majority voting.

| Method | # Sample | MATH | GSM8k | ProofNet | Trips | Meetings | Calendar | StrQA | SciQA | ID | OOD | Overall |
|---|---|---|---|---|---|---|---|---|---|---|---|---|
| | | \multicolumn{3}{c}{Math (ID)} | | \multicolumn{3}{c}{Planning (OOD)} | | \multicolumn{2}{c}{QA (OOD)} | \multicolumn{3}{c}{Average} | | |
| *Pass@N or Larger LLM* | | | | | | | | | | | | |
| Qwen3-8B pass@1 (Lower Bound) | - | 92.4 | 95.6 | 74.1 | 8.1 | 5.5 | 23.5 | 86.8 | 92.7 | 87.4 | 43.3 | 59.8 |
| Qwen3-8B pass@$N$ (Upper Bound) | - | 99.3 | 99.2 | 97.8 | 36.2 | 16.0 | 60.5 | 98.3 | 99.3 | 98.8 | 56.1 | 72.1 |
| Qwen3-14B pass@1 | - | 93.4 | 97.6 | 76.0 | 38.7 | 5.0 | 40.5 | 91.9 | 95.6 | 89.0 | 54.3 | 67.3 |
| *Unsupervised Uncertainty Quantification (UQ)* | | | | | | | | | | | | |
| Majority Voting | - | – | 97.6 | – | – | – | – | 86.6 | 92.5 | – | – | – |
| Min Number of Steps | - | 91.0 | 95.1 | 73.9 | 23.4 | 7.0 | 22.5 | 85.9 | 96.3 | – | – | – |
| MaxProb | - | 92.4 | 96.2 | 76.0 | 20.6 | 5.5 | 31.0 | 84.1 | 96.0 | 88.2 | 47.4 | 62.7 |
| MaxEntropy | - | 92.0 | 96.1 | 74.1 | 6.9 | 4.0 | 29.5 | 86.6 | 95.6 | 87.4 | 44.5 | 60.6 |
| Perplexity | - | 92.7 | 96.4 | 74.1 | 6.9 | 3.5 | 27.5 | 85.6 | 94.5 | 87.7 | 43.6 | 60.1 |
| *PRMs 150× Larger than UHeads* | | | | | | | | | | | | |
| Skywork-PRM-1.5B | Unk | 94.4 | 97.6 | 76.5 | 6.9 | 6.0 | 23.0 | 86.6 | 96.3 | 89.5 | 43.8 | 60.9 |
| H4-Qwen2.5-PRM-1.5B-0.2 | 369K | 91.7 | 95.1 | 71.4 | 15.6 | 4.5 | 22.0 | 84.6 | 94.7 | 86.1 | 44.3 | 59.9 |
| *PRMs 750× to 810× Larger than UHeads* | | | | | | | | | | | | |
| Math-Shepherd-PRM-7B | 440K | 93.0 | 95.5 | 72.8 | 9.1 | 4.0 | 30.0 | 87.3 | 95.8 | 87.1 | 45.2 | 60.9 |
| RLHFlow-PRM-Deepseek-Data | 253K | 92.7 | 96.4 | 71.7 | 8.7 | 3.5 | 25.5 | 87.6 | 94.1 | 86.9 | 43.9 | 60.0 |
| RLHFlow-PRM-Mistral-Data | 273K | 93.7 | 96.3 | 71.7 | 8.4 | 3.5 | 30.0 | 87.8 | 94.1 | 87.2 | 44.8 | 60.7 |
| Universal-PRM-Qwen2.5-Math-7B | 690K | 95.7 | 97.5 | 76.0 | 9.7 | 4.0 | 24.0 | 87.8 | 97.1 | 89.7 | 44.5 | 61.5 |
| Qwen2.5-Math-7B-PRM800k | 263K | 92.7 | 97.3 | 74.4 | 5.9 | 6.0 | 27.5 | 87.1 | 96.9 | 88.1 | 44.7 | 61.0 |
| Qwen2.5-Math-PRM-7B | 860K | 93.7 | 97.8 | 76.0 | 7.2 | 5.5 | 26.5 | 88.1 | 96.9 | 89.2 | 44.8 | 61.5 |
| *Uncertainty Heads (UHeads)* | | | | | | | | | | | | |
| ★ UHead Self-anno (Ours) | 32K | 94.4 | 97.5 | 73.6 | 9.4 | 6.5 | 31.0 | 88.6 | 97.1 | 88.5 | 46.5 | 62.3 |
| ★ UHead DeepSeek-anno (Ours) | 32K | 92.7 | 97.8 | 76.5 | 17.2 | 7.0 | 26.0 | 88.6 | 96.9 | 89.0 | 47.1 | 62.8 |

Table 4: Online best-of-$N$ decoding accuracy across datasets (Qwen3-8B).

| Method | # Sample | MATH | GSM8k | ProofNet | Trips | Meetings | Calendar | StrQA | SciQA | ID | OOD | Overall |
|---|---|---|---|---|---|---|---|---|---|---|---|---|
| | | \multicolumn{3}{c}{Math (ID)} | | \multicolumn{3}{c}{Planning (OOD)} | | \multicolumn{2}{c}{QA (OOD)} | \multicolumn{3}{c}{Average} | | |
| *PRMs 750× to 810× Larger than UHeads* | | | | | | | | | | | | |
| Math-Shepherd-PRM-7B | 440K | 72.9 | 93.7 | 44.7 | 5.3 | 7.7 | 27.5 | 65.0 | 63.4 | 70.43 | 33.78 | 47.53 |
| RLHFlow-PRM-Deepseek-Data | 253K | 74.0 | 94.4 | 48.2 | 5.1 | 8.3 | 28.1 | 66.8 | 63.2 | 72.20 | 34.30 | 48.51 |
| RLHFlow-PRM-Mistral-Data | 273K | 74.1 | 94.4 | 46.9 | 7.5 | 10.6 | 29.6 | 65.0 | 62.4 | 71.80 | 35.02 | 48.81 |
| Universal-PRM-Qwen2.5-Math-7B | 690K | 70.4 | 91.6 | 43.8 | 9.1 | 6.5 | 28.5 | 60.4 | 61.0 | 68.60 | 33.10 | 46.41 |
| Qwen2.5-Math-7B-PRM800k | 263K | 76.4 | 91.3 | 48.0 | 5.1 | 11.5 | 24.5 | 63.8 | 63.2 | 71.90 | 33.62 | 47.98 |
| Qwen2.5-Math-PRM-7B | 860K | 74.0 | 92.7 | 55.9 | 5.0 | 6.6 | 26.0 | 57.6 | 63.8 | 74.20 | 31.80 | 47.70 |
| *Uncertainty Heads (UHeads)* | | | | | | | | | | | | |
| ★ UHead Self-anno (Ours) | 32K | 74.2 | 92.9 | 48.0 | 9.5 | 10.1 | 29.0 | 69.6 | 59.8 | 71.70 | 35.60 | 49.14 |
| ★ UHead DeepSeek-anno (Ours) | 32K | 74.8 | 93.7 | 51.5 | 12.2 | 13.7 | 24.6 | 69.9 | 58.8 | 73.33 | 35.84 | 49.90 |

**Combining PRM with UHead.** We train a logistic regression model that takes as input both the PRM score and the UHead score to predict step correctness. The model is trained on a random subset of 200 questions from the training set. Table 5 reports the step-level PR-AUC, showing that this combination yields additional improvements. These findings suggest that the integration of uncertainty-based and PRM-based signals is a promising direction for future work.

Table 5: Step-level PR-AUC for the experiment combining PRMs with UHead (Qwen-3 8B).

| Method | MATH | GSM8k | ProofNet |
|---|---|---|---|
| PRM1 (Qwen2.5-Math-7B-PRM800k) | .586 | .613 | .301 |
| PRM2 (Qwen2.5-Math-7B) | .531 | .702 | .310 |
| UHead DeepSeek-anno | .529 | .594 | .260 |
| UHead + PRM1 | .613 | .674 | .318 |
| UHead + PRM2 | .573 | .710 | .327 |

**Impact of data quantity and diversity.** Figure 2 (top) shows that larger training sets, more questions and reasoning trajectories, consistently improve UHead performance on both ID and OOD tasks. In practice, however, high-quality prompts are often scarce, making it difficult to scale by prompt expansion alone. An alternative is to sample multiple reasoning trajectories per question. The bottom row of Figure 2 compares scaling strategies, demonstrating that trajectory sampling is an effective way to boost UHead performance.

To examine the effect of data diversity, we train UHeads on two subsets of 2K questions from the training set: one highly similar and one highly diverse. Questions are embedded using Qwen3-Embedding-8B (Zhang et al., 2025b). The similar subset is formed by selecting the 2K nearest

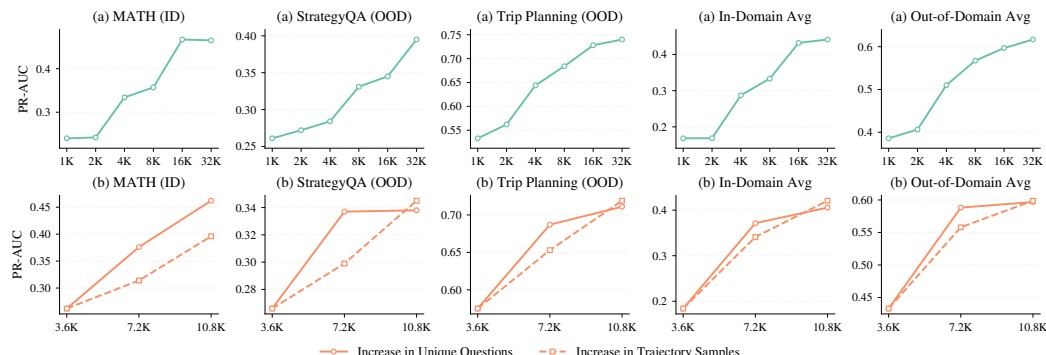

Figure 2: Top row: PR-AUC of UHeads with increasing training set size (x-axis). Bottom row: scaling training data either by adding new unique questions or by sampling additional trajectories. One dataset per domain is shown; other tasks are covered in Appendix A.

neighbors to the median embedding, while the diverse subset is constructed via farthest-first traversal (Gonzalez, 1985). Results in Table 8 show that data diversity benefits UHead performance.

**Scaling $N$.** In Figure 3, we present the best-of-$N$ performance on GSM8k and ScienceQA for different values $1 \leq N \leq 10$. On GSM8k, performance improves with larger $N$ for almost all methods, with both UHead variants consistently achieving the best results across most settings, outperforming the PRM-based baselines. On ScienceQA, accuracy also increases as $N$ grows, but the differences between methods are smaller, and all approaches perform comparably.

## 6 RELATED WORK

**Process reward models.** Research in PRMs has advanced by scaling and refining step-level annotations: from manual labeling (Uesato et al., 2022; Lightman et al., 2023), to MC-based automatic labeling (Wang et al., 2024a; Luo et al., 2024), and consensus methods combining LLM-as-a-Judge and MC estimation (Zhang et al., 2025c; Zhao et al., 2025). Generative PRMs further extend step-level judgment with long CoT (Xiong et al., 2025; Zhao et al., 2025). In contrast, our approach focuses on cost-efficient step-level verification via LLM internal states.

**Uncertainty quantification** methods recently have been employed for test-time scaling and improving reasoning performance (Mo & Xin, 2024; Yin et al., 2024; Zhang et al., 2025a; Fu et al., 2025a; Yan et al., 2025; Kang et al., 2025). However, so far, only weak unsupervised UQ methods have been used. In this work, we propose data-driven UQ techniques that achieve much better performance.

**Formal verification** has recently been used to verify LLM reasoning steps (Zhou et al., 2024; Hu et al., 2025; Liu et al., 2025; Zhou & Zhang, 2025). However, these methods often require specialized autoformalization data for training and are limited to narrow domains (e.g., math proofs). In contrast, we focus on UQ methods and demonstrate their ability to generalize across domains.

## 7 CONCLUSION

We introduced UHead, a lightweight step-level verifier that reads an LLM's own internal states to detect incorrect reasoning. UHead can be trained fully self-supervised without human labels, verifiable final answers, or Monte-Carlo rollouts. Across math, planning, and QA, UHead delivers strong in- and out-of-domain results and is competitive with, or better than, far larger PRMs, making it a practical building block for resource-efficient reasoning systems. Beyond replacing PRMs, UHead complements them: combining PRM scores with UHead's score consistently improves performance, indicating the two signals capture complementary aspects of reasoning quality. This synergy suggests a promising path toward strong hybrid verifiers that marry introspective uncertainty with process rewards. Looking ahead, our findings pave the way for more efficient test-time scaling and self-verification for LLMs in reasoning tasks.

## ETHICS STATEMENT

We use publicly available datasets (MATH, GSM8K, ProofNet, ScienceQA, and StrategyQA) which have no data privacy issues. All artifacts we use are under licenses allowing research usage. Human annotations were conducted by the authors of this paper. We do not identify any other ethical risks associated with this study.

## REPRODUCIBILITY STATEMENT

We will fully open-source our trained UHeads, code, prompts, human annotations, and processed datasets to ensure full reproducibility. For all training, evaluation, and sampling, we fix random seeds to 1 or 42 (see specified in scripts). One major challenge of reproducing exact numbers in our tables from scratch is the use of API-based DeepSeek-R1. API-based LLMs are known to be inherently non-deterministic even if fixing prompts and temperature. To address this, we provide all DeepSeek-R1 annotations used in training and evaluation, allowing others to faithfully reproduce our results. If reproducing from scratch, our codebase also guarantees to reproduce similar trends and observations, even if there are slight differences in exact numbers.

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

# A  ADDITIONAL EXPERIMENTAL RESULTS

To verify our framework works for models of different size, family, and post-training, we conduct step-level correctness and offline BoN evaluation on a Phi-4 UHead trained on Qwen3-8B annotated training data. The step-level correctness prediction and Offline BoN results are presented in Table 6 and Table 7 correspondingly. For budget reason, we do not perform DeepSeek-R1 annotation for Phi-4 training data. On step-level correctness, Qwen3-8B-annotated UHead achieves the best performance in Meeting and Calendar Planning, and the best average performance on OOD tasks. On StrategyQA and overall average, it ranks the second.

For offline BoN, the Qwen3-8B–annotated UHead outperforms two strong Qwen2.5-Math PRMs on MATH and matches the strongest PRMs on GSM8K. On ProofNet and StrategyQA, it also outperforms some much larger PRMs.

Table 6: PR-AUC for detecting incorrect reasoning steps for Phi-4. Best scores are shown in **bold**, and other competitive scores are underlined. # Sample indicates the number of training samples, where each sample corresponds to a reasoning trajectory with step-level annotations. ‡ Qwen2.5-Math-PRM-7B's training data is filtered from an 860K-sample dataset; the exact size after filtering is not specified in their paper.

| Method | # Sample | Math (ID) | | | Planning (OOD) | | | QA (OOD) | | Average | | |
| --- | --- | --- | --- | --- | --- | --- | --- | --- | --- | --- | --- | --- |
| | | MATH | GSM8k | ProofNet | Trips | Meetings | Calendar | StrQA | SciQA | ID Avg. | OOD Avg. | Overall |
| *Unsupervised Uncertainty Quantification (UQ)* | | | | | | | | | | | | |
| Random | - | .106 | .038 | .082 | .552 | .463 | .324 | .172 | .086 | .075 | .319 | .228 |
| MaxProb | - | .127 | .084 | .123 | .618 | .548 | .380 | .252 | .158 | .111 | .391 | .286 |
| MaxEntropy | - | .112 | .079 | .107 | .585 | .533 | .362 | .248 | .135 | .099 | .373 | .270 |
| Perplexity | - | .117 | .066 | .099 | .557 | .508 | .323 | .228 | .143 | .094 | .352 | .255 |
| *PRMs 150× Larger than UHeads* | | | | | | | | | | | | |
| Skywork-PRM-1.5B | Unk | .219 | .181 | .185 | .408 | .467 | .327 | .237 | .415 | .195 | .371 | .305 |
| H4-Qwen2.5-PRM-1.5B-0.2 | 369K | .174 | .061 | .105 | .534 | .476 | .434 | .212 | .116 | .113 | .354 | .264 |
| *PRMs 750× to 810× Larger than UHeads* | | | | | | | | | | | | |
| Math-Shepherd-PRM-7B | 440K | .248 | .188 | .188 | .747 | .584 | .489 | .249 | .327 | .208 | .479 | .378 |
| RLHFlow-PRM-Deepseek-8B | 253K | .200 | .263 | .109 | .558 | .455 | .343 | .315 | **.440** | .191 | .422 | .335 |
| RLHFlow-PRM-Mistral-8B | 273K | .141 | .195 | .093 | .462 | .424 | .309 | .261 | .311 | .143 | .353 | .274 |
| Universal-PRM-Qwen2.5-Math-7B | 690K | **.485** | .213 | **.263** | .741 | .559 | .497 | .320 | .252 | .320 | .474 | .416 |
| Qwen2.5-Math-7B-PRM800k | 265K | .474 | **.406** | .238 | **.825** | .599 | .568 | **.355** | .329 | **.373** | .535 | **.474** |
| Qwen2.5-Math-PRM-7B | 860K | .427 | .377 | .240 | .791 | .594 | .555 | .333 | .310 | .348 | .517 | .453 |
| *Uncertainty Heads (UHeads)* | | | | | | | | | | | | |
| ★ UHead Qwen3-8B-anno (Ours) | **32K** | .404 | .340 | .155 | .756 | **.646** | **.592** | .347 | .347 | .300 | **.538** | .448 |

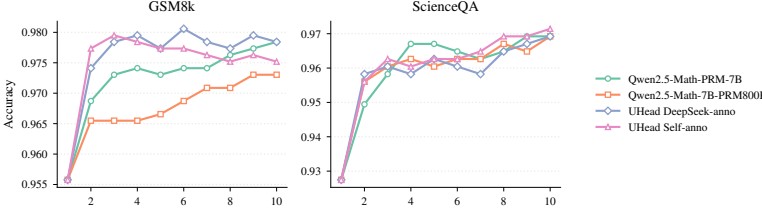

Figure 3: Best-of-$N$ performance on GSM8k and ScienceQA for different $N$ values (Qwen3-8B).

Table 7: Offline Best-of-$N$ decoding accuracy across datasets for Phi-4 model. For datasets with verifiable final answer, we also provide Majority Voting. We use max uncertainty aggregation over steps.

| Method | # Sample | Math (In-domain) | | | Reasoning QA | |
|---|---|---|---|---|---|---|
| | | MATH | GSM8k | ProofNet | StrQA | SciQA |
| *Pass@N* | | | | | | |
| Phi-4 pass@1 (Lower Bound) | - | 90.5 | 97.9 | 86.8 | 93.0 | 95.7 |
| Phi-4 pass@$N$ (Upper Bound) | - | 98.2 | 100. | 97.2 | 99.1 | 99.8 |
| *Unsupervised Uncertainty Quantification (UQ)* | | | | | | |
| Majority Voting | - | – | **100.** | – | 93.2 | 94.6 |
| Min Number of Steps | - | 91.1 | **100.** | 91.0 | 93.9 | 97.1 |
| MaxProb | - | 92.3 | 98.9 | 88.2 | 94.1 | 98.0 |
| MaxEntropy | - | 94.1 | 98.4 | 87.5 | 93.0 | 98.0 |
| Perplexity | - | 92.9 | 98.9 | 87.5 | 93.4 | **98.4** |
| *PRMs 150× Larger than UHeads* | | | | | | |
| Skywork-PRM-1.5B | Unk | 95.9 | 99.5 | **95.7** | 92.5 | 96.6 |
| H4-Qwen2.5-PRM-1.5B-0.2 | 369K | 93.5 | **100.** | 89.2 | 91.8 | 96.4 |
| *PRMs 750× to 810× Larger than UHeads* | | | | | | |
| Math-Shepherd-PRM-7B | 440K | 92.9 | 98.4 | 95.2 | 93.2 | 96.6 |
| RLHFlow-PRM-Deepseek-Data | 253K | 94.1 | 98.9 | 89.6 | **94.8** | 98.0 |
| RLHFlow-PRM-Mistral-Data | 273K | 92.3 | 98.4 | 90.3 | 94.1 | 97.3 |
| Universal-PRM-Qwen2.5-Math-7B | 690K | **96.4** | **100.** | 93.5 | 92.5 | 96.8 |
| Qwen2.5-Math-7B-PRM800k | 263K | 93.5 | **100.** | 92.4 | 93.9 | 97.5 |
| Qwen2.5-Math-PRM-7B | 860K | 93.5 | **100.** | 94.4 | 94.1 | 97.5 |
| *Uncertainty Heads (UHeads)* | | | | | | |
| ★ UHead Qwen3-8B-anno (Ours) | **32K** | 95.9 | **100.** | 93.1 | 93.0 | 95.7 |

Table 8: PR-AUC performance of UHeads trained on the most Diverse/Indiverse 2K questions (3 trajectories sampled for each question). Data diversity benefits the overall performance of UHead.

| Method | # Sample | Math (ID) | | | Planning (OOD) | | | QA (OOD) | | Average | | |
|---|---|---|---|---|---|---|---|---|---|---|---|---|
| | | MATH | GSM8k | ProofNet | Trips | Meetings | Calendar | StrQA | SciQA | ID Avg. | OOD Avg. | Overall Avg. |
| UHead DeepSeek-anno Indiverse | 6K | .308 | .549 | **.205** | .626 | .687 | .685 | **.377** | .251 | .354 | .525 | .461 |
| UHead DeepSeek-anno Diverse | 6K | **.409** | **.575** | .180 | **.707** | **.793** | **.792** | .271 | **.325** | **.388** | **.578** | **.507** |

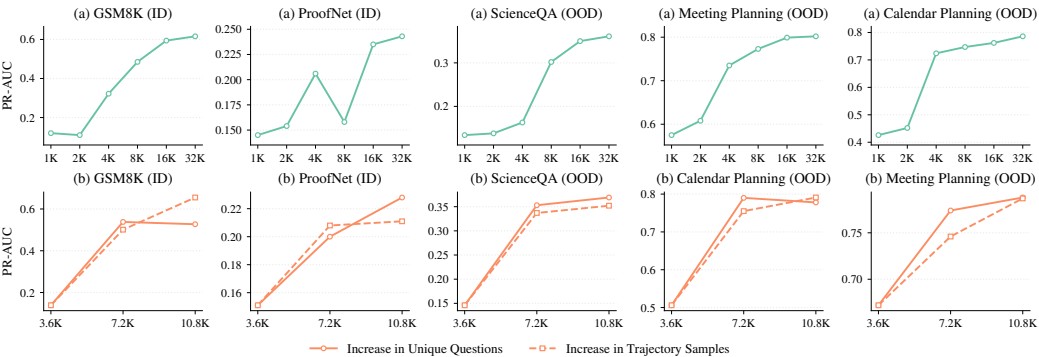

Figure 4: Top row: PR-AUC of UHeads with increasing training set size (x-axis). Bottom row: scaling training data either by adding new unique questions or by sampling additional trajectories. One dataset per domain is shown.

## B    DISCUSSION AND LIMITATIONS

§5 shows that UHead performance increases with larger training data and benefits from data diversity. Curves in Figure 2 and Figure 4, although show diminishing marginal gains, do not seem to reach the top for tasks like StrategyQA. Therefore, it seems possible to further unleash the potential of UHead with further data scaling – sampling more reasoning trajectories per question or adding new questions beyond the PRM800K training set. In this work, we do not involve questions outside the PRM800K training set to establish a fair comparison with PRMs trained on data derived from this set (e.g., Qwen2.5-Math-7B-PRM800K). Due to the high cost of DeepSeek-R1 (annotation of 32K reasoning trajectories cost >2000 USD), we also do not annotate more reasoning trajectories per question. We leave this promising scaling to future work, including the integration of diverse, high-quality questions outside the math domain. To save annotation cost, future work may also substitute API-based LLM with recent small but capable reasoning models, for example, GPT-OSS (Agarwal et al., 2025).

Another limitation regarding the UHead framework is that it needs to be trained on the internal signal of target LLMs it supervises. Once trained, a UHead cannot be directly applied to another LLM, since it depends on model-specific internal states. By contrast, PRMs can supervise any LLM because they only operate on output tokens. However, UHeads are highly efficient in parameter size and training data, making their training relatively inexpensive. Once trained, UHeads can significantly save inference costs compared to PRMs. In practice, applications may focus on a small number of widely used models. If UHeads for these models are shared online (e.g., on Hugging-Face), practitioners could readily download and apply them without training from scratch. Future work may also investigate fine-tuning UHeads to adapt to customized or fine-tuned versions of target LLMs.

## C    PROMPTS

### C.1    LLM PROMPT

We use a domain-agnostic, format-enforcing prompt to elicit structured step-by-step reasoning, detailed in Figure 5.

### C.2    ANNOTATION PROMPTS

For step-level annotation, we use the 2-stage prompts shown in Figure 6, while chain-level correctness annotations are obtained using the prompt in Figure 7.

```
<|im_start|>user
You will be presented with a <Question>. Before providing the [Answer
    ], you should first think step-by-step carefully.

Your response format:
<start of response>
Reasoning Steps:
- Step 1: [Your first reasoning step]
- Step 2: [Your second reasoning step]
- Step 3: [Next step, and so on...]
...
- Step N: [Final reasoning step]
<Answer>: [Your final answer]
<end of response>

Strict Requirements:
- DO NOT include any text outside the specified format.
- Each reasoning step MUST be written on a **single line only**: NO
    line breaks, bullet points, or substeps within a step.
- Each step should express one precise and **self-contained** logical
    operation, deduction, calculation, or fact application.
- Steps MUST provide explicit result of the step or concrete reasoning
     outcomes. Avoid vague explanations or meta-descriptions of the
    reasoning process.
  - For example:
     - Good: "- Step 1: Multiply 5 by 4, which equals 20."
     - Bad: "- Step 1: Multiply 5 by 4." (no result of the step or
         concrete reasoning outcome)
- Continue writing steps until the problem is solved.
- Violating ANY requirement above is NOT acceptable.

Now answer:
<Question>: {q}<|im_end|>
<|im_start|>assistant
<think>

</think>
Reasoning Steps:
```

Figure 5: Prompt template used to elicit structured step-by-step reasoning from the model.

**Step-Level Annotation – Stage 1 Prompt:**

```
You are given a problem, a ground-truth solution, and a step-by-
    step student solution. Your task is to analyze each step in the
     student's solution to determine whether it is both logically
    correct and relevant.

Instructions:
- Carefully examine each student step for logical errors or
    unnecessary/redundant reasoning.
- If all steps are correct and they lead to the same final answer
    as the ground-truth solution, conclude that there are no errors
    .
- If any step contains an error that would prevent the student from
     reaching the correct solution, identify and report those
    specific steps with an explanation.

PROBLEM:
{problem}

GROUND-TRUTH SOLUTION:
{answer}

STUDENT'S SOLUTION STEPS:
{steps}

Now, please evaluate whether the student's steps are correct and
    logical.
```

**Step-Level Annotation – Stage 2 Prompt (Postprocessing):**

```
You are given:
- A problem
- A student's step-by-step solution (as a Python list of string
    steps)
- An assessment of student's solution

Your task:
Output a single Python list where each element is:
- 1 if the corresponding step is correct
- 0 if the step is incorrect

Important:
- Output only the list, nothing else.
- The list must have the same length as the number of steps.

PROBLEM:
{problem}

STUDENT'S SOLUTION STEPS:
{steps}

ASSESSMENT OF STUDENT SOLUTION STEPS:
{reply}

OUTPUT LIST:
```

Figure 6: Two-step prompting procedure for step-level correctness annotation. The first stage evaluates the solution and identifies flaws, while the second converts this into binary correctness labels.

```
You will be given a <Problem> and its proposed <Solution>. Your
    task is to assess whether the solution is **correct** or **
    incorrect**.

Respond using the **exact format** below, do not include any text
    outside this template.
Output format:
<start of response>
Solution comments:
... your comments on the solution, explaining reasoning, pointing
    out any errors or confirming correctness ...
<Grade>: (Correct|Incorrect)
<end of response>

<Problem>: {problem}

<Solution>: {solution}
```

Figure 7: Prompt used for annotating chain-level correctness by evaluating the full reasoning trace.

# D  TRAINING DETAILS

## D.1  TRAINING DATA ANNOTATION DETAILS

In our experiments, we employ two LLM judges for the annotation of UHead training data. For DeepSeek-R1, we follow the officially recommended inference setting, using a temperature of 0.6. Similarly, when using Qwen3-8B as training data annotator, we also follow the recommended inference hyperparameter, using a temperature of 0.7, a top_k of 20, and a top_p of 0.95. We access Qwen3-8B through vLLM local deployment, and access DeepSeek-R1 through the DeepSeek API. The annotation prompts are detailed in Figure 6. We do not set max tokens to restrict the length of reasoning chains.

## D.2  HYPERPARAMETER DETAILS

For all experiments on `Phi-4` and `Qwen3-8B`, we use the same set of hyperparameters. We use a learning rate of 5e-4, a batch size of 128, and a positive class weight of 3. For UHeads, we use one layer of transformer encoder with a hidden size of 512 and 16 attention heads. All training continues for 5 epochs. We use 10% of the training data and a 200-sample set of GSM8K, ScienceQA, and StrategyQA for validation and best checkpoint selection.

## D.3  COMPUTATION DETAILS

All experiments are carried out on cluster nodes with 4 GH200 GPUs and another cluster node with 2 H100 GPUs.

# E  TEST DATASET DETAILS

The mathematical domain includes MATH (high school and competition-style math problems; Hendrycks et al., 2021), GSM8K (grade-school math word problems; Cobbe et al., 2021), and ProofNet (natural language proofs of undergraduate-level math problems; Azerbayev et al., 2023). Planning domain includes NaturalPlan (Zheng et al., 2024) (spans three real-world tasks – trip planning, meeting planning, and calendar scheduling). The QA domain includes ScienceQA without multi-modal context (Lu et al., 2022) (covers 26 science subjects from elementary to high school) and StrategyQA (Geva et al., 2021) – a general knowledge QA benchmark that requires implicit multi-step reasoning. Due to compute limitations, we evaluate on subsets of the test sets.

Table 9 presents the statistics of the test datasets used for the step-level benchmark. Importantly, we rely on DeepSeek-R1 for step-level evaluation, which is an expensive API-based LLM. We thus take a subset of test sets for this evaluation. While only having a few hundred questions, there is a substantial number of reasoning steps, which allows us to draw representative insights.

Table 9: Test dataset statistics used for step-level benchmark.

| Dataset | Model | #Questions | #Steps | Mean #Steps | %Correct Steps | Answer Mean Len (Tokens) |
|---|---|---|---|---|---|---|
| MATH | Qwen3-8B | 200 | 1203 | 6.2 | 85.5% | 204 |
| | Phi-4 | 200 | 2296 | 11.5 | 89.2% | 417 |
| GSM8k | Qwen3-8B | 200 | 1056 | 5.1 | 94.0% | 139 |
| | Phi-4 | 200 | 1286 | 6.4 | 95.7% | 174 |
| ProofNet | Qwen3-8B | 186 | 1211 | 6.5 | 81.9% | 236 |
| | Phi-4 | 186 | 2227 | 12.0 | 91.1% | 515 |
| Trips | Qwen3-8B | 320 | 5492 | 17.2 | 46.9% | 641 |
| | Phi-4 | 320 | 4486 | 14.0 | 44.87% | 584 |
| Meetings | Qwen3-8B | 200 | 2569 | 12.8 | 40.9% | 739 |
| | Phi-4 | 200 | 3350 | 16.75 | 52.3% | 584 |
| Calendar | Qwen3-8B | 200 | 1616 | 8.1 | 52.5% | 365 |
| | Phi-4 | 200 | 2172 | 10.86 | 66.9% | 428 |
| StrategyQA | Qwen3-8B | 500 | 2865 | 5.73 | 81.1% | 121 |
| | Phi-4 | 500 | 3288 | 6.6 | 82.6% | 182 |
| ScienceQA | Qwen3-8B | 500 | 2621 | 5.24 | 88.7% | 110 |
| | Phi-4 | 500 | 3415 | 6.8 | 91.9% | 172 |

# F DATASET EXAMPLES

| Dataset | Question | Answer |
|---------|----------|--------|
| GSM8k | Natalia sold clips to 48 of her friends in April, and then she sold half as many clips in May. How many clips did Natalia sell altogether in April and May? | - Step 1: Natalia sold 48 clips in April.
- Step 2: Natalia sold half as many clips in May, which is $48 \div 2 = 24$.
- Step 3: Add them: $48 + 24 = 72$.
<Answer>: 72 |
| ProofNet | If $r$ is rational ($r \neq 0$) and $x$ is irrational, prove that $r + x$ is irrational. | - Step 1: Assume for contradiction that $r+x$ is rational.
- Step 2: Then $x = (r + x) - r$ is rational.
- Step 3: This contradicts $x$ being irrational.
- Step 4: Therefore $r + x$ is irrational.
<Answer>: $r + x$ is irrational |
| MATH | Between what two consecutive integers is $\sqrt{30}+\sqrt{50}$ located? Answer in the form ''m and n''. | - Step 1: $\sqrt{30} \approx 5.477$.
- Step 2: $\sqrt{50} \approx 7.071$.
- Step 3: Sum $\approx 12.548$.
- Step 4: Lies between 12 and 13.
<Answer>: 12 and 13 |
| Trips | You plan to visit 3 European cities for 14 days. Stay in Istanbul for 5 days, Tallinn for 4 days, Zurich for 7 days. From day 1-7, there is a show in Zurich. Direct flights exist: Istanbul--Tallinn, Zurich--Tallinn, Zurich--Istanbul. Find a valid trip plan. | - Step 1: Start in Zurich for days 1-7 (show).
- Step 2: Fly Zurich → Tallinn, stay days 8-11.
- Step 3: Fly Tallinn → Istanbul, stay days 12-14.
<Answer>: Day 1-7: Zurich, Day 8-11: Tallinn, Day 12-14: Istanbul |
| Meetings | You arrive at Financial District at 9:00AM. Richard: Marina District 3:30-5:30PM (need 90 min). Andrew: Alamo Square 5:00-10:00PM (need 120 min). Travel: FD-Marina 15min, FD-Alamo 17min, Marina-Alamo 15min. Plan a schedule to meet both. | - Step 1: Leave FD 2:45PM, arrive Marina 3:00PM.
- Step 2: Meet Richard 3:30-5:00PM.
- Step 3: Leave Marina 5:15PM, arrive Alamo 5:30PM.
- Step 4: Meet Andrew 5:30-7:30PM.
<Answer>: FD → Marina → Alamo schedule works |
| Calendar | Schedule a 30min meeting for Laura and Paul on Monday between 9:00-17:00. Laura busy: 11:30-12:30, 14:30-15:00, 16:00-16:30. Paul busy: 9:30-10:00, 11:00-14:30, 15:00-17:00. Paul prefers not after 9:30. | - Step 1: Laura free 9:00-11:30.
- Step 2: Paul free 9:00-9:30 and before 9:30 only.
- Step 3: Intersection is 9:00-9:30.
<Answer>: Meeting 9:00-9:30 |
| StrategyQA | Did Donatello use a smartphone? | - Step 1: Donatello lived 1386-1466.
- Step 2: Smartphones invented 21st century (iPhone 2007).
- Step 3: Impossible for him to have used one.
<Answer>: No, he did not |
| ScienceQA | Which word is not like the others?
A. horse
B. goat
C. squirrel
D. leg | - Step 1: Horse, goat, squirrel are animals.
- Step 2: Leg is a body part.
- Step 3: Outlier is D.
<Answer>: D. leg |

Table 10: Examples of question-answer pairs from each dataset, ordered by domain: Math (GSM8k, ProofNet, MATH), Planning (Trips, Meetings, Calendar), and QA (StrategyQA, ScienceQA).

Table 10 presents examples of questions and LLM answers from the training datasets.

## G   HUMAN VALIDATION DETAILS

We first randomly shuffle the DeepSeek-R1 annotated test set. The human annotation then starts from the first row of the shuffled dataset. Human annotators are not provided with DeepSeek-R1's rationales (i.e., contents within the `<think>` tag) and must independently assess the correctness of each reasoning step, following the step-level correctness definition in Figure 6.

The annotation difficulty varies by tasks. ScienceQA is the easiest, which mostly requires common sense and usually takes 1 to 2 minutes to annotate one step. StrategyQA requires extensive fact-checking against Wikipedia, but since the dataset provides relevant facts for grading (available to both human and LLM annotators),annotation usually takes around 2 minutes per step. ProofNet and three planning datasets are the most demanding, often requiring more than 5 minutes per step due to the reasoning length and question complexity.

Regardless of dataset difficulty, annotators are requested to annotate at least 100 steps from each dataset, with the option to annotate more depending on their availability.

## H   THE USAGE OF LLMS

In this paper, we mainly use LLMs as objects of study. We also use DeepSeek-R1 and Qwen3-8B as training data annotators, as detailed in §3 and Appendix C.1. We also use DeepSeek-R1 as a judge to grade answer correctness, as detailed in §4. DeepSeek-R1's accuracy in these tasks is manually validated through human annotation (see Table 1). In coding and writing, we use LLM assistants (e.g., ChatGPT) to identify grammar errors and debug. Such usage is under careful human supervision.

