# OpenReview forum: "Reasoning with Confidence: Efficient Verification of LLM Reasoning Steps via Uncertainty Heads"
_ICLR.cc/2026/Conference — ICLR 2026 Conference Withdrawn Submission_

### Official Review · Reviewer_mWf8 · 2025-10-28

**Soundness:** 2
**Presentation:** 2
**Contribution:** 2
**Rating:** 2
**Confidence:** 3

**Summary:**

This paper addresses the challenge of verifying intermediate reasoning step correctness in LLMs’ multi-step reasoning and proposes a lightweight Uncertainty Quantification Head (UHead) to replace computationally expensive Process Reward Models (PRMs).

**Strengths:**

- The proposed method is compared with comprehensive baselines.
- Process reward is important for the development of Large reasoning models.

**Weaknesses:**

1. The proposed method is not clear, how is U (r(j)t ∣r(j)<t , x) be estimated, what's the archecture of the U-heads.
2. It seems that this paper utilizes the U-head to learn the uncertainty for process-reward estimation. Since the U-head is from another work, what's the contribution of this work?
3.The method should be evaluated on the latest PRM benchmark like PRMBench

**Questions:**

U-head contains few parameters compared with LLMs, but does it rely on the embedding or hidden states of LLMs? If In that case, we can not say that U-head is an more efficient methon than some simple baseline like LLM-as-judge.

---

> ### Author Response · Authors · 2025-11-23
>
> We thank Reviewer mWf8 for acknowledging the comprehensiveness of our experiment, and the importance of our research question. Here we address the concerns (Cs) and answer the Questions (Qs).
>
> ---
>
> **C1. The proposed method is not clear, how is U (r(j)t | r(j)<t , x) be estimated, what's the archecture of the U-heads.**
>
> The UHead architecture is based on a simple lightweight 1-2-layer Transformer. We begin by projecting the feature space into a latent space that matches the dimensionality required by the Transformer subnetwork. The Transformer then produces contextualized token representations, which we average across tokens for the reasoning step. Finally, a 2-layer classification module, regularized with dropout, generates the final uncertainty score.
>
> ---
>
> **C2. It seems that this paper utilizes the U-head to learn the uncertainty for process-reward estimation. Since the U-head is from another work, what's the contribution of this work? **
>
> Our contribution is a **new paradigm** for reasoning step-verification, which supervises the LLM internal states using UHead, rather than only reading final texts with PRMs. Our work advance the field in following dimensions (see also lines 102 to 114):
>
> - Prior work in UQ mainly focuses on **factuality** (e.g., Einstein was born in Bern. Factually correct or not?), including the previous UHead paper. We are the first to adapt supervised uncertainty estimation to **step-level reasoning verification**, and the first to **systematically compare** uncertainty-based methods (e.g., popular self-uncertainty) with PRMs
>
> - To fulfil exploration on how to adapt supervised UQ for reasoning, we contribute a correctness-based automatic data annotation pipeline. Existing PRMs all require large-scale data of strict quality filtering, while our data creation pipeline is much more efficient.
>
> - We are the first to prove that supervised UQ can be efficient and generalizable alternatives for reasoning step verification, through comprehensive evaluations in ID, OOD, test-time scaling with BoN, and step-level correctness. See tables 2-4.
>
> ---
>
> **C3. The method should be evaluated on the latest PRM benchmark like PRMBench**
>
> We respectfully disagree. Because UHeads rely on internal states of the target LLM, they can only supervise on-policy reasoning traces generated by that same model. Consequently, off-policy traces, such as those in PRMBench, cannot be evaluated by UHeads, which is shown by prior work [1].
>
> Given this structural limitation, prestigious off-policy benchmarks like PRMBench are not applicable for step-level correctness evaluation. To compensate, we provide comprehensive test-time scaling evaluations (Tables 3-4), which offer a robust and complementary assessment of UHead behavior.
>
> Thanks for raising this concern! We will include this discussion regarding PRMBench in the final version.
>
> [1] A Head to Predict and Ahead to Question: Pre-trained Uncertainty Quantification Heads for Hallucination Detection in LLM Outputs, 2025. URL: https://arxiv.org/abs/2505.08200
>
> ---
>
> **Q1: U-head contains few parameters compared with LLMs, but does it rely on the embedding or hidden states of LLMs?**
>
> UHead does rely on the internal states of the base LLM, but our efficiency claims refer to **the marginal cost of uncertainty estimation after the LLM has already generated its reasoning steps**. In the intended use case, the user first runs the LLM to produce the chain of thought; only then do they apply an uncertainty method (LLM-as-judge, PRM, or UHead). At this point, the LLM forward pass is already complete, and UHead simply uses the cached internal activations, which come “for free” relative to the original generation. Therefore, the only additional computation is the lightweight UHead itself, whose cost is determined solely by its parameter count. This makes UHead substantially more efficient than larger PRMs or LLM-as-judge methods that require a full additional model inference.

---

### Official Review · Reviewer_4SUw · 2025-11-01

**Soundness:** 2
**Presentation:** 3
**Contribution:** 1
**Rating:** 2
**Confidence:** 4

**Summary:**

The authors use model uncertainty as process supervision to guide the model’s reasoning steps. To perform uncertainty estimation, they train a lightweight value head in a supervised manner to predict the model’s uncertainty. The training data are labeled either by the model itself serving as the supervisory model or by a third-party supervisory model.

The authors conduct experiments on mathematics, planning, and QA datasets, comparing their approach with several unsupervised uncertainty estimation methods and third-party process reward models.

**Strengths:**

1. The method proposed by the authors is lightweight — it only requires training a value head, which makes it highly efficient.
2. The authors proposed an automated training data construction scheme.
3. The authors conducted extensive experiments, comparing their approach across datasets from three different domains.

**Weaknesses:**

1. The proposed method lacks novelty, as many prior works have already trained process reward models (PRMs) or used uncertainty estimation as a supervision signal. For example, the baseline methods cited by the authors employ similar ideas. The main contribution of this paper is merely implementing such supervision through a lightweight value head. And UHead is also an existing work.
2. The authors' definition of uncertainty lacks rigor. Generally speaking, a metric trained directly from accuracy should not be regarded as a measure of uncertainty. For example, when a model produces a particular wrong answer very frequently during random sampling, its uncertainty about this that answer should be very low. However, under the training method proposed by the authors, such a case would yield a high uncertainty value. For the definition of uncertainty, I recommend reading this paper: https://arxiv.org/pdf/1802.10501

**Questions:**

Have you compared the results between full-parameter fine-tuning and Uhead?

---

> ### Author Response · Authors · 2025-11-23
>
> We thank Reviewer 4SUw for acknowledging the efficiency of our reasoning step verification paradigm and our data annotation pipeline, and the comprehensiveness of our experiments. Here we respond the concerns (C) and answer the questions (Q):
>
> ---
>
> **C1: The proposed method lacks novelty, as many prior works have already trained process reward models (PRMs) or used uncertainty estimation as a supervision signal.**
>
> PRMs leverage only textual generations, whereas our approach leverages internal states of LLMs  without even looking at textual generations. This makes it  fundamentally different from conventional PRMs.
>
> At the same time, our approach is different from recent works that start applying simplistic unsupervised uncertainty quantification methods. As shown in our experiments, unsupervised uncertainty-based test time scaling falls significantly behind PRMs and our method based on UHead.
>
> Summarizing, our approach is conceptually different both from PRMs and methods based on unsupervised uncertainty quantification. It combines the strong generalization properties characteristic of UQ-based approaches with the performance competitive with PRMs, while offering massive computational advantages. UHeads require activating only ~10M parameters instead of billions, since they operate solely on cached internal states. This yields a new, highly efficient, and more generalizable paradigm for step-level reasoning verification.
>
> ---
>
> **C2: The authors' definition of uncertainty lacks rigor, a metric trained directly from accuracy should not be regarded as a measure of uncertainty.**
>
> Thank you for the comment! In fact, some recent works frame UQ as the problem of approximating expected risk (Kotelevskii et al., 2025; Hofman et al., 2024). Following this definition, any proper approximation of the risk can be viewed as some type of uncertainty. Kotelevskii et al., 2025, for example, connect various UQ methods suggested in the literature to various proper scoring rules for risk approximation. Supervised hallucination detectors trained to approximate accuracy are not different in this sense as they still give some estimation of the expected risk. Our main motivation to connect scores from supervised hallucination detectors with uncertainty is to highlight that we can efficiently obtain some credibility score for reasoning steps by probing internal states of the LLM itself, without using external knowledge sources during inference (such as huge PRMs).
>
> **C3: For example, when a model produces a particular wrong answer very frequently during random sampling, its uncertainty about this that answer should be very low. However, under the training method proposed by the authors, such a case would yield a high uncertainty value.**
>
> If the model constantly produces a wrong answer, then it has low aleatoric uncertainty. However, it does not mean that epistemic uncertainty should be low in this case. Since the model answers incorrectly, it does not have enough knowledge to answer the question, this means high epistemic uncertainty. In practice, usually we need predictive uncertainty (the sum of aleatoric and epistemic), which our method aims to approximate.
>
> ---
>
> **Q1: Have you compared the results between full-parameter fine-tuning and Uhead?**
>
>
> **This question might be based on a wrong premise** because UHead is **not** a parameter-efficient tuning (PEFT) algorithm like Adaptor or LoRA.
>
> - **PEFT**
>   - **Goal:** Change the base LLM’s behavior toward a favored direction.
>   - **At Training Time:** Parameter-efficient since the base LLM is frozen.
>   - **At Inference Time:** No extra efficiency.
>     - Both the base LLM and PEFT parameters must be activated, **or** LoRA must be merged into the base LLM.
>     - For CoT supervision, an extra PRM is required.
>       **Parameter count:** base LLM parameters + PRM parameters.
>
> - **UHead**
>   - **Goal:** Supervise LLM CoT **without modifying** its behavior.
>   - **At Training Time:** Parameter-efficient since the base LLM is frozen.
>   - **At Inference Time:** Extra efficiency.
>     - UHead only reads the base LLM’s activations.
>       **Parameter count:** base LLM parameters + UHead parameters.
>
> In summary, UHead is not integrated into the base model like PEFT approaches, nor does it modify the model’s parameters. Therefore, a full-parameter finetuning comparison is not applicable. UHeads operate independently on cached activations.
>
> ---
>
> **References:**
>
> Kotelevskii, N., Kondratyev, V., Takáč, M., Moulines, É., & Panov, M. (2025). From Risk to Uncertainty: Generating Predictive Uncertainty Measures via Bayesian Estimation. In Proceedings of the Thirteenth International Conference on Learning Representations (ICLR 2025), Singapore.
> Hofman, Paul, Yusuf Sale, and Eyke Hüllermeier. "Quantifying aleatoric and epistemic uncertainty with proper scoring rules." arXiv preprint arXiv:2404.12215 (2024).

---

### Official Review · Reviewer_sfZj · 2025-11-02

**Soundness:** 3
**Presentation:** 2
**Contribution:** 2
**Rating:** 6
**Confidence:** 4

**Summary:**

This paper introduces UHeads (Uncertainty quantification Heads), a lightweight alternative to Process Reward Models (PRMs) for verifying step-level correctness in LLM reasoning chains. UHeads are small transformer modules (<10M parameters) trained on frozen LLM internal states to predict step-level uncertainty, with labels generated either by larger models (DeepSeek-R1) or through self-supervision. The authors demonstrate that despite being 750-810× smaller than PRMs, UHeads achieve competitive performance across mathematics, planning, and QA tasks, particularly excelling in out-of-domain scenarios, suggesting that LLMs' internal states encode meaningful uncertainty signals for reasoning verification.

**Strengths:**

1.  The proposed UHeads achieve comparable or superior performance to PRMs while using 750-810× fewer parameters (9.8M vs 7-8B), offering a highly efficient alternative for step-level reasoning verification that significantly reduces inference costs and memory requirements.
2. UHeads demonstrate superior generalization capabilities, particularly on OOD tasks where they consistently outperform much larger PRMs, suggesting they capture more transferable uncertainty signals rather than overfitting to domain-specific patterns.
3. The automatic annotation pipeline eliminates requirements for human labels, verifiable final answers, or costly Monte Carlo rollouts, supporting both external supervision (via DeepSeek-R1) and self-supervision approaches with comparable performance.

**Weaknesses:**

1. Tables 2-4 and 6 consistently show UHeads underperforming strong PRM baselines on in-domain mathematical tasks (MATH, GSM8K), with gaps of 5-10% in PR-AUC, raising questions about whether the computational savings justify the accuracy trade-off for domain-specific applications.
2. The 256-token generation limit during training data creation may constrain the method's applicability to more complex reasoning tasks like AIME problems that require tens of thousands of tokens, potentially limiting the approach's generalizability.
3. Given that UHeads require training on specific LLM internal states while PRMs can be used off-the-shelf across different models, and considering the performance gaps on certain tasks (e.g., ScienceQA where RLHFlow-PRM-DeepSeek significantly outperforms), the overall value proposition compared to training a single general-purpose PRM remains unclear.

**Questions:**

1. How does the approach handle step boundary definition in complex reasoning chains that include self-verification, backtracking, or recursive refinement? The paper's reliance on structured prompts may not generalize to more naturalistic reasoning patterns.
2. In Section 2.3, the notation P(y|x,D) appears problematic since training on data D fundamentally changes model parameters θ rather than just conditioning the distribution. Should this be reformulated as P_θ'(y|x) where θ' represents post-training parameters?
3. What training factors contribute to UHeads' underperformance on in-domain tasks compared to PRMs? A deeper analysis of failure modes and potential improvements would strengthen the paper's contribution.

---

> ### Author Response · Authors · 2025-11-23
>
> We thank Reviewer sfZj for acknowledging the UHead-based stepwise verification paradigm as “highly efficient” and having “superior generalization”, the efficiency of our data creation pipeline and its effectiveness in supporting external/self-annotation. We address the concerns (C) and answer the questions (Q):
>
> ---
>
> **C1: Tables 2-4 and 6 consistently show UHeads underperforming PRM on in-domain mathematical tasks (MATH, GSM8K)**
>
> We respectfully disagree with the “consistent underperforming". UHead outperforms PRM on GSM8K and ProofNet in Table 3, and achieves second best in many other in-domain tasks in Table 2 and 4. Considering the model capacity (10M vs. 7-8B) and PRM’s heavy optimization on ID tasks, UHeads’ performance is already very satisfactory. Moreover, UHead outperforms smaller PRMs (1.5B parameters) across all datasets, although those models are still 150× larger.
>
> OOD generalization is another advantage of UHead. Therefore, we argue that the commented performance-efficiency trade-off does not exist. **UHead is good on both**.
>
> ---
>
> **C2: The 256-token generation limit during training data creation may constrain the method's applicability to more complex reasoning tasks.**
>
> **UHeads can supervise longer reasoning chains than the training data length restriction**. Due to budget reasons, we limited the length of training data. However, **for testing, we have no restriction on generation length**. Take planning as an example, which is a complex reasoning task. For trip, calendar, and meeting plan, UHeads are tested on generations with average length of **640, 365, and 739 tokens**, which is far longer than the upper bound of training data **256** tokens.
>
> ---
>
> **C3: UHeads require training on specific LLM internal states [...] The overall value proposition compared to training a single general-purpose PRM remains unclear.**
>
> We agree on the advantages of a single general-purpose PRM, but **UHead can be preferred in settings where efficiency and generalizability are important** (also see lines 79-99).
>
> - **Efficiency**. At inference time, UHeads activate orders of magnitude fewer parameters than PRMs, making them substantially more efficient. While UHeads do require lightweight model-specific training, this cost is minimal due to their small size and data efficiency (e.g., only 32K LLM-annotated samples in our setup).
>
> - **Generalization**. Across Tables 2-4, UHeads outperform PRMs that are ~800× larger on many OOD tasks, demonstrating strong generalization. They are not universally superior, for example, they underperform RLHFlow-Llama-8B on ScienceQA (Table 2), but they still outperform PRMs in most settings, and even on ScienceQA they achieve the best offline BoN score (Table 3).
>
> ---
>
> **Q1: The paper's reliance on structured prompts may not generalize to more naturalistic reasoning patterns.**
>
> Thank you for raising this concern. Our goal in this paper is to provide a **proof-of-concept** that LLM internal states can be used to supervise reasoning more efficiently and more generalizable than external PRMs. To cleanly evaluate this question, we deliberately avoid the additional complexity of segmenting naturalistic reasoning traces, which may involve recursion, self-verification, and backtracking, into semantically coherent steps.
>
> Instead, we use structured prompts to obtain already “segemented” CoT. This simplification is crucial: since no prior work trains supervised modules to estimate uncertainty over reasoning steps, it allows us to (i) establish a clear experimental foundation for step-level correctness (Table 2) and (ii) study test-time scaling effects (Tables 3–4). Without this controlled setup, even the basic validation of step-level correctness (Table 1) would be infeasible.
>
> Exploring UHeads under fully naturalistic, unconstrained reasoning patterns is an interesting next step, but firmly **beyond the scope of this initial study.**

---

> > ### Author Response · Authors · 2025-11-23
> >
> > ---
> >
> > **Q2: In Section 2.3, the notation P(y|x,D) appears problematic since training on data D fundamentally changes model parameters θ rather than just conditioning the distribution. Should this be reformulated as P_θ'(y|x) where θ' represents post-training parameters?**
> >
> > In the Bayesian approach to modelling the parameters are represented by a prior distribution P(θ). Training conditions prior on the data, which results in a posterior distribution of parameters P(θ|D). Θ' designates a particular sample from posterior. In this particular case, we reason in terms of uncertainty using information theoretical and Bayesian framework, and are interested in predictive posterior distribution for all possible parameters θ without specifying particular samples of parameters, so this formula is correct. In practice, we can compute approximations through using some specific post-training parameters, yet we would need multiple variations of Θ': (ensemble, Monte Carlo dropout, Bayesian layers, etc.)
> >
> > ---
> >
> > **Q3: What training factors contribute to UHeads' underperformance on in-domain tasks compared to PRMs? A deeper analysis of failure modes and potential improvements would strengthen the paper's contribution**
> >
> > Thanks for raising this but we respectfully argue that UHeads’ **in-domain performance is already strong**. In Table 3, UHeads outperform ALL 800x larger PRMs GSM8K and ProofNet. In other ID columns of table 2-4, UHeads’ performance still beats many PRMs that are 750-800x larger. It also constantly outperforms 150x larger PRMs.
> >
> > We did a qualitative analysis by carefully inspecting samples where UHead predicts high uncertainty and Qwen2.5-7B-PRM predicts low, and vice versa. Unfortunately, we did not recognize a significant pattern of UHead-PRM behavior difference.

---

### Official Review · Reviewer_DYzG · 2025-11-04

**Soundness:** 2
**Presentation:** 1
**Contribution:** 2
**Rating:** 2
**Confidence:** 4

**Summary:**

This paper introduces a method for step-wise verification based on quantifying the uncertainty involved in the reward predictions for each reasoning step. It implements a UHead, a classification module on top of LLM’s hidden states and uses its predictions for scoring verification rewards. Empirically, the paper provides experiments in step-level correctness and offline/online best-of-N using verifier-guided search, claiming on par performance with 7B-8B PRM models and strong OOD generalization.

**Strengths:**

- The idea of using UQ methods for unsupervised/self-supervised verification is well motivated and justified.

- The paper brings several baselines, particularly on UQ methods for verification, which is a good benchmarking for UQ-based model-based verification research.

- The paper brings an interesting OOD evaluation setting, which is very relevant but overlooked in PRM research.

**Weaknesses:**

- The major concern in the paper is its clarity/presentation on describing the proposed methodology. The background section brings a section about UQ but does not follow up on it when describing the method in Section 3. The core technique in the paper is the UHead, but this is not formally described in the paper, making it not self-contained. There are no details on how uncertainty is estimated or how/whether the terms in the equation of Section 2.3 are computed, nor what is the nature of the uncertainty estimated (e.g., predictive, epistemic).

- From the description provided in Section 3, the UHead seems to be a classification network on top of a base LLM hidden state, and the uncertainty here relates to the softmax entropy among Yes/No classes. If this understanding is correct, then there are a few points to consider:
    - It would be important to compare against the predictive entropy from the LLM itself, i.e., consider the (re-normalized) Yes/No distribution conditioned on the reasoning step and compute its entropy as score; this baseline will validate if the classification training is indeed needed.
    - The claims about “comparing” against models that are 150x, 810x times larger sounds misleading since the method still requires inference over a base LLM to extract features, so all the parameters of the base LLM are also activated in the process of generating verification. This should be counted as well if the goal is to compare model sizes.
    - There are also strong claims about the UHeads being general, plug and play, and that they “generalize across tasks, languages and domains”. These claims are unclear and unjustified. From the paper description, these are classification models trained on top of self-supervision or even DeepSeek-R1 labels, so we need proper evidence to support these claims, otherwise I would expect them to behave similarly to other Adapter models in the literature.

- The Related Work section is very superficial. It provides a little of contextualization and does not contrast with other similar works. There is also recent work in uncertainty-aware step-wise verification missed, e.g., [1, 2].

- The paper does not report confidence intervals to assess statistical significance in the results. In fact, the paper does not mention how many experimental seeds were used (I assume it is a single one). Prior literature has raised how sensitive math reasoning benchmarking is for small changes [3], requiring a more statistical grounding to evaluate whether the reported takeaways are meaningful or just observation noise.
    - As an illustrative example, Figure 3 (left) is used as evidence to claim scaling improvements for the proposed method. The reported gap in performance is less than 1% of accuracy (over Qwen2.5-Math-PRM-7B), which diminishes as N increases. There is no way to assess statistical significance here, yet the paper claims the “consistently better results”. The same lack of statistical rigor extends to all reported experiments, which makes it hard to evaluate scientific claims.

Overall, I believe the paper requires a good rewriting in the methodological section to improve clarity on the proposed method. Some of the claims (as described above) needs to be calibrated, and the experiments should report performance across different seeds with proper confidence intervals. The related work section may also be polished to better contextualize with the literature and contrast with similar methods.


References:

[1] Cao et. al. More bang for the buck: Process reward modeling with entropy-driven uncertainty, 2025.

[2] Ye et. al. Uncertainty-Aware Step-wise Verification with Generative Reward Models, 2025.

[3] Hochlehnert et. al. A Sober Look at Progress in Language Model Reasoning: Pitfalls and Paths to Reproducibility. COLM, 2025.

**Questions:**

N/A

---

> ### Author Response · Authors · 2025-11-23
>
> We thank the reviewer for acknowledging our idea as “well-motivated and justified”, our effort of covering comprehensive baselines and OOD evaluations. Here we address the concerns (Cs) one by one:
>
> ---
>
> **Q1. The core technique in the paper is the UHead, but this is not formally described in the paper, making it not self-contained. There are no details on how uncertainty is estimated or how/whether the terms in the equation of Section 2.3 are computed, nor what is the nature of the uncertainty estimated (e.g., predictive, epistemic).**
>
> We thank the reviewer for the helpful suggestion. We have revised Sections 2.3 and 3 to include a more thorough description of the UHead architecture.
>
> The UHead architecture is based on a simple lightweight 1-2-layer Transformer. We begin by projecting the feature space into a latent space that matches the dimensionality required by the Transformer subnetwork. The Transformer then produces contextualized token representations, which we average across tokens for the reasoning step. Finally, a 2-layer classification module, regularized with dropout, generates the final uncertainty score. UHead directly approximates the predictive uncertainty.
>
> ---
>
> **C2: ”[...] It would be important to compare against predictive entropy from the LLM from re-normalized Yes/No distribution. [...]”**
>
> We believe the requested baselines (e.g., entropy from the LLM’s yes/no distribution and other uncertainty quantification methods) **are already in paper**. Among the unsupervised UQ baselines in Table 2, we already include the **P(True) method**, which directly queries the LLM about each reasoning step using a True/False format and then uses the model’s probability of the “True” token. While this is different from computing entropy over a normalized Yes/No distribution, we believe it captures a comparable judgment from the model. As shown in Table 2, UHead consistently outperforms P(True) as well as all other (10 in total) unsupervised baselines, indicating that leveraging internal representations provides a stronger signal than relying solely on output-token probabilities.
>
> In table 2, we also include many other strong UQ baselines, including self-certainty which shows great performance in reasoning path selection. In summary, these baselines would already prove the effectiveness of our proposed method.
>
> ---
>
> **C3: ”The claims about “comparing” against models that are 150x, 810x times larger sounds misleading since the method still requires inference over a base LLM to extract features.”**
>
> We respectfully argue that **our comparison is valid**. Both PRMs and UHead rely on the same base language model to generate the reasoning steps (e.g. an 8B LLM). However, our comparison focuses on the **additional computation** required to perform uncertainty estimation after the reasoning steps have been generated." Specifically:
>
> 1) PRM: The base LLM generates the CoTs (8B params activated), followed by a separate 8B parameter PRM that activates all 8B parameters to grade each step.
>
> 2) UHead: As in case of 1) The base LLM generates the CoT (8B params activated) with the internal states cached. Then UHead predicts uncertainty from the already-cached internal states. Only the UHead parameters (~10M) are activated during this stage.
>
> Here, the 800x times comparison is for the last step, 8B vs. 10M.
>
> You also mentioned that **“otherwise I would expect them to behave similarly to other Adapter models in the literature”**. UHead and Adapter differ fundamentally. **Adapters must activate the base LLM and directly influence its generation**, whereas **UHeads operate solely on cached internal states, fully decoupled from base-model forward passes.**
>
> Thanks for pointing this out! We will improve the presentation clarity regarding how UHeads work and its efficiency.
>
> ---
>
> **C4: There are also strong claims about the UHeads being general, plug and play, and that they “generalize across tasks, languages and domains”.**
>
> We’d like to point out that The generalization of UHead across tasks and domains is **supported by comprehensive experiments in Table 2, 3, and 4**.
>
> Importantly, this claim in line 205 **does not describe** our own contribution; rather, it summarizes the findings of prior work [1], which demonstrated that UHeads generalize effectively across multiple dimensions **for factual-error detection**. Building on this established result, our contribution is to extend the UHead architecture to **reasoning** and to assess whether similar generalization properties hold. We did not evaluate cross-lingual generalization in our work, and we do not claim otherwise.
>
> [1] A Head to Predict and Ahead to Question: Pre-trained Uncertainty Quantification Heads for Hallucination Detection in LLM Outputs, 2025.

---

> > ### Author Response · Authors · 2025-11-23
> >
> > ---
> >
> >
> > **C5: The Related Work section is very superficial.**
> >
> > Thanks for raising this point! Here we extend the related work following your suggestion to better compare our contributions to the previous work:
> >
> >
> > - **PRM.** Research in PRMs has advanced by scaling and refining step-level annotations … [original lines 480-464] … (Xiong et al., 2025; Zhao et al., 2025). Cao et al. (2025) leverage anchor token uncertainty in PRM training data sampling to improve data diversity, efficiency and quality. Ye et al. (2025) quantify PRM’s uncertainty to directly improve PRM’s accuracy and calibration. In our work, we introduce UHead as a new paradigm for step-level reasoning supervision. Unlike PRMs monitoring CoT texts, UHeads monitor LLM internal states and achieve good efficiency and generalizability.
> >
> > In short, compared to the PRM literature, our work introduces a novel step-level supervision paradigm by monitoring LLM internal states. Compared to work that leverages uncertainty quantification for reasoning, we are the first proposing data-driven techniques that achieve much better performance.
> >
> > ---
> >
> > **C6: The paper does not report confidence intervals to assess statistical significance in the results. In fact, the paper does not mention how many experimental seeds were used.**
> >
> > We calculated bootstrapping std for 2 selected PRMs, 2 UHeads and some unsupervised UQ baselines on 2 qa datasets for step-level benchmark in Table 2:
> >
> > | Method                     | StrQA              | SciQA              |
> > |----------------------------|--------------------|--------------------|
> > | Random                     | 0.116 ± 0.006      | 0.125 ± 0.007      |
> > | MaxProb                    | 0.114 ± 0.009      | 0.259 ± 0.010      |
> > | MaxEntropy                 | 0.119 ± 0.008      | 0.227 ± 0.011      |
> > | Perplexity                 | 0.110 ± 0.007      | 0.228 ± 0.009      |
> > | UHead (Self-anno)          | 0.394 ± 0.013      | 0.401 ± 0.013      |
> > | UHead (Deepseek-anno)      | 0.395 ± 0.013      | 0.361 ± 0.013      |
> > | Qwen2.5-Math-7B-PRM800k    | 0.362 ± 0.013      | 0.404 ± 0.013      |
> > | Qwen2.5-Math-PRM-7B        | 0.334 ± 0.014      | 0.429 ± 0.014      |
> >
> >
> > The results show that the improvements introduced by UHead are statistically significant.
> >
> > We use a single fixed seed for all experiments. We will clarify this in the final version of the paper for reproducibility.

---

### Author Response · Authors · 2025-12-03
**Summarization of the Discussion Period**

Dear AC, SAC, and Program Committee,

We thank the reviewers for their careful evaluation of our work. Building on their comments and the additional experiments and clarifications added during the rebuttal, we summarize below the main strengths of the submission and how we address the key concerns.

---
Below we briefly summarize key strengths highlighted in the reviews:

- A **lightweight** UHead-based verifier operating on frozen LLM internal states, using <10M parameters and providing a **highly efficient** alternative to large PRMs for step-level reasoning verification (noted by sfZj, 4SUw).

- **Strong empirical evaluation and generalization**: comprehensive baselines and experiments across mathematics, planning, and QA, with carefully designed OOD settings where UHeads often outperform much larger PRMs (noted by DYzG, sfZj, 4SUw, mWf8).

- A fully **automatic data creation pipeline** and well-motivated use of unsupervised/self-supervised uncertainty signals, avoiding human labels, verifiable final answers, or costly rollouts (noted by DYzG, sfZj, 4SUw).

---

Some of the raised concerns and our responses to them are:

**Q1. Lack of methodological clarity (UHead architecture, how uncertainty is estimated)**

We expanded Sections 2.3 and 3 to provide a formal, self-contained description of UHead (projection → 1-2-layer Transformer → pooled step representation → 2-layer classifier). We clarified that UHeads approximate *predictive* uncertainty and explained the Bayesian notation.

---

**Q2. Missing baseline of entropy over Yes/No distribution**

The requested baseline is already represented by *P(True)* and other unsupervised UQ baselines in Table 2. UHeads consistently outperform all 10 unsupervised UQ metrics, demonstrating the advantage of using internal states.

---

**Q3. Comparisons vs 150× to 800× larger PRMs may be misleading since both rely on the same base LLM.**

The comparison refers to *additional* parameters activated for uncertainty estimation. PRMs require a full 7-8B forward pass; UHeads use only ~10M parameters on cached activations. We also clarified that UHeads are not adapters and never re-activate the base model.

---

**Q4. Claims about generality (tasks, domains, languages) not fully supported.**

We clarified that cross-lingual generalization is **not** claimed. The statement referenced prior UHead work on factuality. Our contribution is extending UHeads to **reasoning** and showing broad task/domain generalization across Tables 2 to 4.

---

**Q5. Superficial related work section.**

We expanded the PRM and UQ discussions, and clarified how our paradigm differs: UHeads supervise **internal states**, while PRM supervises text and is less efficient than UHeads.

---

**Q6. No confidence intervals; unclear number of seeds.**

We added bootstrapped std for key UHead and PRM baselines on two datasets, showing statistical significance. All experiments use a fixed seed.

---

**Q7. UHeads underperform PRMs on in-domain (ID) math tasks.**

We show that UHeads outperform all 800× larger PRMs on GSM8K and ProofNet (Table 3), and outperform 150× larger PRMs across datasets. Their ID performance is competitive and often second-best. Superior OOD performance is another highlight of UHeads

---


**Q8. 256-token training limit may prevent handling long reasoning chains.**

Test-time has no length restriction. In planning tasks, UHeads operate on chains averaging 365-739 tokens, far exceeding the 256-token training cap.

---

**Q9. UHead value unclear vs a single general-purpose PRM**

UHeads offer (i) orders-of-magnitude lower inference cost (10M vs 7-8B params activated), (ii) strong OOD generalization, and (iii) minimal training cost (32K samples). This efficiency-generalization performance is not achievable with PRMs.

---

**Q10. Structured prompts may not reflect natural reasoning; limited scope.**
Structured prompting provides controlled step segmentation necessary for the first proof-of-concept of internal-state supervision. Studying unconstrained reasoning is future work.

---

**Q11. Missing evaluation on PRMBench.**

PRMBench is **off-policy**, while UHeads require **on-policy internal states** of the target LLM. Thus PRMBench is structurally incompatible. We instead provide extensive ID/OOD and BoN scaling evaluations. Moreover, PRMBench uses MATH questions, which are covered in our test suite.

---

**Q12. How does UHead compare to full-parameter or PEFT finetuning?**

This comparison is not applicable because UHeads do **NOT** modify or merge with the base model. They operate independently on cached activations and are not PEFT methods.

---

We hope this structured summary helps the new AC quickly grasp the key concerns raised by reviewers, our corresponding responses, and the additional evidence provided during the discussion period.


All the best,

Author Team

---

### Note · Authors · 2026-01-06

I have read and agree with the venue's withdrawal policy on behalf of myself and my co-authors.